# Exospheric Solar Wind Model Based on Regularized Kappa Distributions for the Electrons Constrained by Parker Solar Probe Observations

Viviane Pierrard [1,2,*], Maximilien Péters de Bonhome [2], Jasper Halekas [3], Charline Audoor [2], Phyllis Whittlesey [4] and Roberto Livi [4]

1   Solar-Terrestrial Center of Excellence and Space Physics, Royal Belgian Institute for Space Aeronomy, B-1180 Brussels, Belgium
2   Center for Space Radiations, TECLIM, Earth and Life Institute (ELI), Université Catholique de Louvain, B-1348 Louvain-La-Neuve, Belgium; maximilien.peters@student.uclouvain.be (M.P.d.B.); charline.audoor@gmail.com (C.A.)
3   Department of Physics and Astronomy, University of Iowa, Iowa City, IA 52242, USA; jasper.halekas@gmail.com
4   Space Sciences Laboratory, University of California, Berkeley, CA 94720, USA; phyllisw@berkeley.edu (P.W.); rlivi@berkeley.edu (R.L.)
*   Correspondence: viviane.pierrard@aeronomie.be

**Abstract:** In the present work, the kinetic exospheric model of the solar wind is improved by considering regularized Kappa distributions that have no diverging moments through consideration of a cut-off at relativistic velocities. The model becomes valid even for kappa indices lower than 2, which is important since low values of kappa are observed in the fast solar wind. The exospheric model shows that the electric potential accelerates the wind to supersonic velocities. The presence of suprathermal Strahl electrons at the exobase can further increase the velocity to higher values, leading to profiles comparable to the observations in the fast and slow wind at all radial distances. The kappa index is not the only parameter that influences the acceleration of the wind: the difference in the altitude of the exobase also makes a significant difference between the fast and slow wind. The exobase is located at lower altitudes in the coronal holes where the density is smaller than in the other regions of the corona, allowing the wind originating from the holes to be accelerated to higher velocities. The new observations of Parker Solar Probe are used to constrain the model. The observations at low radial distances show suprathermal electrons already well present in the Strahl in the antisunward direction and a deficit in the sunward direction, confirming the exospheric feature of almost no incoming particles. For proton distributions, we observe that the proton tail parallel to the magnetic field is already present at 17.2 Rs.

**Keywords:** solar wind; exospheric models; regularized kappa; Parker Solar Probe; suprathermal electrons; velocity distributions; acceleration; solar corona; space plasmas

## 1. Introduction

The new observations of the Parker Solar Probe (PSP) launched in August 2018 provide a unique opportunity to study the radial evolution of the solar wind from its formation close to the Sun to the heliosphere and to compare with predictions from models. This paper aims to explore the physical mechanisms responsible for the heating of the corona and the acceleration of the solar wind, which remain hot topics of research.

The first successful thermally driven solar wind model [1] predicted bulk velocities of several hundred km/s even before any measurement could be made in the interplanetary space. Since the plasma is already collisionless at low radial distances from the Sun, Lemaire and Scherer [2] argued that the collision-dominated assumptions used in the Parker's model are not valid above the exobase, located at low altitude (below 6 solar radii (Rs)) in the solar atmosphere. They used the exospheric approach to model the solar expansion, using

Maxwellian distributions for the particles as boundary conditions at low radial distances. In exospheric models, the quasi-neutrality and zero current conditions lead to the formation of an ambipolar electric potential that accelerates the protons and other positive ions to supersonic velocities. The slow wind is well reproduced, but like in magnetohydrodynamic (MHD) models, unrealistic higher temperatures or other energy sources have to be assumed in the coronal holes to reproduce the high speed solar wind. Parker [3] showed that the MHD approach used in his model is equivalent to the exospheric approach in the limit of negligible electron mass.

The electron distributions in the solar wind near 1 AU are observed to be non-Maxwellian: they have a thermal (~10 eV) Maxwellian core but also a much hotter (~50 eV) halo (~5–10% of the density) whose velocity distribution fits a Kappa distribution, and a highly anisotropic, antisunward field-aligned Strahl that results from the escape of thermal electrons from the hot solar corona and carries heat flux outward (e.g., [4]). Helios' observations indicated that the isotropic halo could be due to scattering of the Strahl [5], and recent PSP observations seem to confirm this paradigm [6].

Following the pioneer works of [7], Pierrard and Lemaire [8] have demonstrated that the presence of an enhanced population of suprathermal particles at a low distance in the corona has the effect of increasing the temperature in planetary and stellar atmospheres in hydrostatic equilibrium. The presence of such suprathermal particles is general in space plasmas, suggesting a universal mechanism for their formation. Using a sum of two Maxwellians with different temperatures, or even better Kappa distributions that well fit observed distributions with power law suprathermal tails, it was shown that even a small enhancement of suprathermal particles is sufficient for the coronal temperature to reach $10^6$ K at 2 Rs when starting from 6000 K at the top of the chromosphere ([9,10] for a review). Moreover, the presence of enhanced suprathermal electron tails accelerates the wind above the exobase in exospheric regions by increasing the electric potential and the flux of escaping electrons [11]. This gives a natural explanation for the fast wind originating from coronal holes, where the density and the temperature are lower than in the other coronal regions [9]. Differential heating and acceleration of minor ions are also predicted using the Kappa exospheric approach, in agreement with ion observations in the solar wind [12]. Even if exospheric models provide a simplified first approximation by considering only the effects of the forces, they show the importance of the electric potential in accelerating the wind, even when considering Maxwellian distributions for the particles at the exobase.

More sophisticated models have also considered the effects of Coulomb collisions, in addition to the external forces, by solving the Fokker-Planck equation at low radial distances [13]. They reproduce distributions with a Maxwellian core, while the suprathermal tails can remain important since the energetic particles are almost not affected by Coulomb collisions. Waves like whistlers [14] and other plasma instabilities [15–17] also help to understand the characteristics of the observed distributions, and especially their temperature anisotropies and heat fluxes. Linear theory suggests that whistler waves propagating in the antisunward direction with an oblique wave vector with respect to the background magnetic field can be driven unstable by the Strahl through the anomalous cyclotron resonance and scatter Strahl electrons in pitch angle through velocity space, leading to the halo component. Nevertheless, PSP observations show that whistlers are very rarely observed inside ~28 Rs (~0.13 AU). Outside 28 Rs, they occur with a plasma beta-heat flux occurrence consistent with the whistler heat flux instability [18].

Several other solar wind acceleration processes have been proposed, mainly involving waves and/or magnetic reconnection ([19–21] and references there in). Even if waves exist and can play a role in the differential heating and acceleration of minor ions, it is unclear if their amplitudes are sufficient to power the wind [22]. The fast solar wind can be generated by extended perpendicular ion heating in coronal holes [23]. The kinetic mechanism responsible for this heating has not yet been determined. One long-standing possibility is the resonant-cyclotron dissipation of ion-cyclotron waves, replenished from

a turbulent cascade of interacting counter-propagating Alfvén waves. In addition to Alfvén wave heating, another candidate mechanism for the still debated energy responsible for accelerating the outflowing plasma to high speeds can be related to the magnetic interchange reconnection (e.g., [24]).

Now that more precise observations are available very close to the Sun with Parker Solar Probe (PSP) (e.g., [25–27]), the solar wind acceleration processes can be analyzed in more detail, leading to improvements in the models since the first in situ solar wind observations in the sixties ([28] for a review).

In the present work, we improve the Kappa exospheric model by considering regularized Kappa distributions [29] so that the power law is no longer limited to kappa values $\kappa > 2$ to calculate the heat flux, contrary to what is imposed when using Olbertian Kappa or modified Kappa [30]. The fits of the observed velocity distributions of the particles by Kappa functions in the solar wind can give kappa indices close and even sometimes lower than 2 (e.g., [10,31] for a review). The regularized Kappa function that has recently been introduced [29,32] takes into account the relativistic effects by imposing a cut-off to the Kappa distribution for velocities reaching the light velocity so that none moment is diverging whatever the value of kappa. Such regularized distributions open many perspectives in plasma physics and are used to improve the exospheric model of the solar wind. This improvement and its consequences are explained in Section 2. In Section 3, the results of model the exospheric model are compared with different solar wind observations at different radial distances, mainly PSP at low distances, Solar Orbiter at intermediate distances, OMNI at 1 AU, and ULYSSES at large distances. Section 4 shows the importance of the electric potential to accelerate the wind and the good agreement with recent PSP observations. Section 5 uses PSP measurements of electron distributions at 17.2 Rs to explore the crucial presence of suprathermal electrons at low distances. Section 6 illustrates simultaneous PSP observations of the proton velocity distribution function (VDF) at 17.2 Rs to determine where the formation of the proton beams occurs. Section 7 shows that the proton velocity-temperature correlation already exists at low radial distances and explores the radial evolution of other correlations. Section 8 provides the conclusions of this study.

## 2. Exospheric Model with Regularized Kappa

The Lorentzian ion exospheric model [8] considered Olbertian Kappa distributions for the electrons, as such functions fit very well the solar wind observed distributions [10,31]. The macroscopic characteristics of the plasmas are then obtained by integrating the VDF f($\mathbf{r}$, $\mathbf{v}$, $t$) on the velocity space. The moments of order $l$ correspond to the integration on the velocity space of the distribution multiplied by the power $l$ of the velocity $v$:

$$M(l) \sim \int_{-\infty}^{\infty} (\mathbf{v} - \mathbf{u})^l f(\mathbf{r}, \mathbf{v}, t) d\mathbf{v} \tag{1}$$

where $\mathbf{v}$ is the velocity vector, $\mathbf{u}$ is the averaged velocity vector, $\mathbf{r}$ is the position vector, and $t$ is the time. The number density n corresponds to the moment of order 0, the flux to the moment of order 1, the pressure to the moment of order 2, and the heat flux to the moment of order 3. The other characteristics, like the bulk velocity, temperatures . . . are also obtained from these definitions (see [8]). Moments of higher orders can also be defined.

With standard Kappa functions, the kappa index had to respect the condition $\kappa > (l + 1)/2$ in order to ensure non-divergent moments of order $l$. For instance, to define the heat flux with the Olbertian truncated Kappa distribution (and with the modified Kappa function as well), the $\kappa$ parameter must be $\kappa > 2$. In the exospheric model, the calculation of the moments starts to diverge for $\kappa \leq 2$ for truncated Kappa distributions and for $\kappa \leq 3/2$ for the temperature of isotropic distributions. To avoid this divergence, it is necessary to take into account the relativistic effects that limit the velocity range of integration to $[-c, c]$ where c is the light velocity. To cut-off the Kappa distribution at very high velocities close

to the light velocity, Scherer et al. [29] introduced the regularized Kappa distribution (RKD), where an exponential factor is added to the standard Kappa in the function:

$$f_R(\kappa, v) = n N_R \left(1 + v^2/(\kappa \, \theta^2)\right)^{-\kappa-1} \exp(-\alpha^2 v^2/\theta^2) \tag{2}$$

where $0 < \alpha < 1$ ($\alpha$ *is the cut-off parameter*), $N_R$ is a normalized constant, and $\theta$ is the thermal velocity of the Olbertian as defined in [10]. They found that $\alpha$ must satisfy $\alpha > \theta/c$ to avoid the contribution to the moments of superluminal particles that have speed values above the speed of light c, so that $\alpha = 0.012$ is an appropriate value for a proper cut-off of the electron solar wind distribution that excludes velocities larger than the speed of light in the vacuum [32].

Scherer et al. [29] define the normalized constant

$$N_R = (\pi \kappa \theta)^{-3/2} \, _{[\,]}\mathscr{U}_{[0]} \, (\kappa, \alpha), \tag{3}$$

by introducing the function $_{[m]}\mathscr{U}_{[n]}(\kappa, \alpha)$ that is a ratio of two Kummer (or Tricomi) functions $U$. This ratio is expressed as follows:

$$_{[m]}\mathscr{U}_{[n]}(\kappa,\alpha) = U((3+m)/2,(3+m)/2 - \kappa, \alpha^2\kappa)/U((3+n)/2,(3+n)/2 - \kappa, \alpha^2\kappa) \tag{4}$$

Table 2 of [33] gives the moments for the isotropic non-drifting distribution function. This table allows us to determine the expression of each moment of the regularized Kappa distribution $f_R(\kappa, \alpha)$ and make the links with the standard Kappa distribution $f_K(\kappa)$. By applying these changes in the initial exospheric model, one can finally adapt it to regularized distributions and obtain the new expressions of the macroscopic quantities valid for any kappa including for $\kappa \leq 2$. For values of $\kappa > 2$, the regularized Kappa distributions give the same moments as the standard Olbertian Kappa. The regularized Kappa also allows anyone to calculate moments of any order $l$ for any value of kappa because there is no divergence anymore for

$$l \geq 2\kappa - 1. \tag{5}$$

For the moments that were not divergent with the standard Kappa, they are not modified by the Regularized Kappa because the cut-off concerns only the very high velocities close to c.

Figure 1 shows the moments from the model based on Regularized Kappa electron distributions for $\kappa$ indices ranging from 1.8 to 2.1.

Figure 1 illustrates the mechanism of the solar wind acceleration explained in the introduction. When the kappa index has a smaller value, the suprathermal tails of the electron distribution are larger, leading to a higher electron flux. It produces a larger polarisation electric field (top right panel) that generates a higher proton flux to keep the solar wind quasi-neutral. The bulk speed is therefore increased (2d row right panel). The model explains why one finds mainly fast winds with a lower kappa index [34]. With an exobase located at a radial distance of 4 Rs and a temperature for electrons and protons of $10^6$ K like in Figure 1, a kappa of 1.8 produces a bulk speed of 1000 km s$^{-1}$, and decreasing the kappa to even lower values increases the bulk velocity to high values. On average, the bulk velocity of the high-speed wind originating from coronal holes at high solar latitudes $> 40°$ was measured by ULYSSES to be around 760 km s$^{-1}$ [35], which is obtained in Figure 1 with a value of kappa close to 2. Moreover, by fitting ULYSSES electron distributions with Kappa functions [31], values of kappa were found to be close to 2, and sometimes lower, even if this is not the most frequent case. The occurrence of $\kappa < 2$ is relatively rare but permits the reproduction of very high-speed wind.

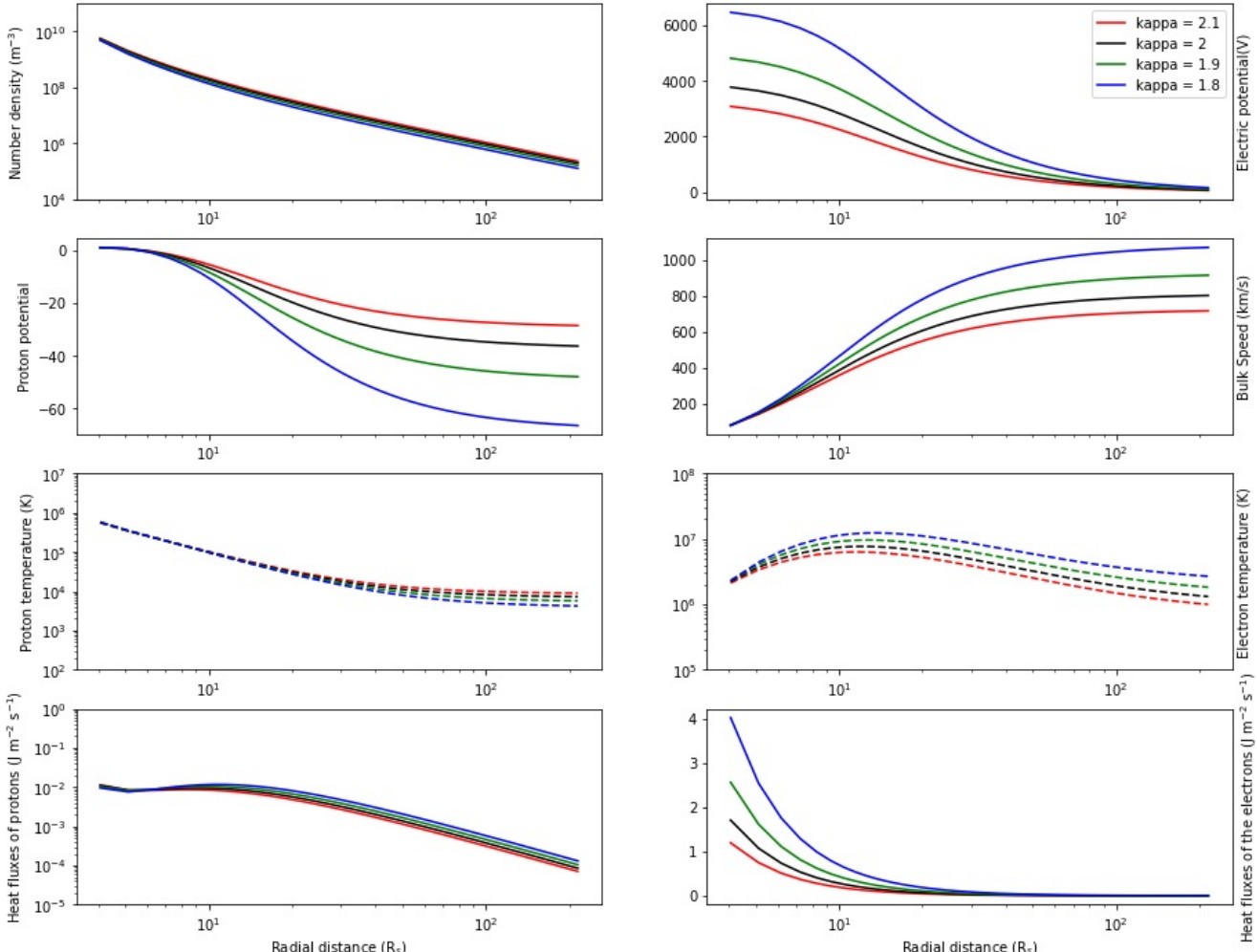

**Figure 1.** The number density (**top left panel**), electric potential (**top right**), proton potential (**2d row, left**), bulk speed (**2d row, right**), proton temperature (**3d row left**), electron temperature (**3d row right**), heat flux of protons (**bottom left**) and of electrons (**bottom right**) calculated by the model for $\kappa$ = 1.8 (blue), 1.9 (green), 2 (black), 2.1 (red). The model took as initial conditions proton and electron temperatures of $10^6$ K, an exobase radial distance of 4 Rs, and goes to a radial distance of 215 Rs (1 AU) corresponding to the Sun-Earth distance.

The highest moments of the protons (temperature, heat flux) do not change significantly with the different $\kappa$ since their VDF is assumed to be a truncated Maxwellian (see Section 5). Its flux and bulk speed are increased, but its Maxwellian distribution remains the same.

One observes that the electron temperature has a maximum of around 10 Rs (3d row, right panel). This maximum is higher when the $\kappa$ index is lower and reaches too high values for very low $\kappa$, at least starting from $10^6$ K at the exobase.

Being the main particles of the solar wind, protons and electrons define the electrostatic potential and have the same bulk speed if the density of the other ions is considered to be negligible. The bulk speed is strongly increasing at low distances before tending towards an approximately constant value. Higher bulk velocities are achieved when the kappa index and/or the exobase are lower. Other ions can also be considered in the Kappa model [36] and thus in the present regularized model. If enhanced populations of energetic particles are present at low altitudes, it was shown that velocity filtration could explain the high temperatures of the heavy ions observed in the corona and, thus, their acceleration in the solar wind with the exospheric model [12].

One obtains that the heat flux increases (bottom panels) with low values of kappa, especially for electrons. Spitzer–Harm conduction [37] corresponds to the heat flux calculated assuming a truncated Maxwellian distribution for the electrons and underestimates what is obtained in the presence of suprathermal electrons below 10 Rs. PSP data show that the heat flux observed in the solar wind at larger distances is actually below the Spizer-Harm prediction, probably due to instabilities [38].

## 3. Comparison between the Model and New Solar Wind Observations

### 3.1. Averaging All Measurements with the Distance

Parker Solar Probe (PSP) and Solar Orbiter (SOLO) are recently launched spacecraft orbiting around the Sun with an elliptical orbit closing in with each revolution with a minimal perihelion that will reach 9.86 Rs in 2024 for PSP and 60 Rs in 2027 for SOLO.

Figure 2 shows a comparison between the exospheric model for values of $\kappa$ of 3, 4, 5, and 8 and the averaged observations during solar minimum of:

| | | |
|---|---|---|
| PSP | from 15 October 2018 | from 0.08 to 0.80 AU, |
| SOLO | from 7 July 2020 | from 0.59 to 0.99 AU, |
| OMNI | from 15 October 2018 | at 1 AU, and |
| ULYSSES (UY) | from 1 January 1995–31 December 1996 | from 1.34–1.36 AU |

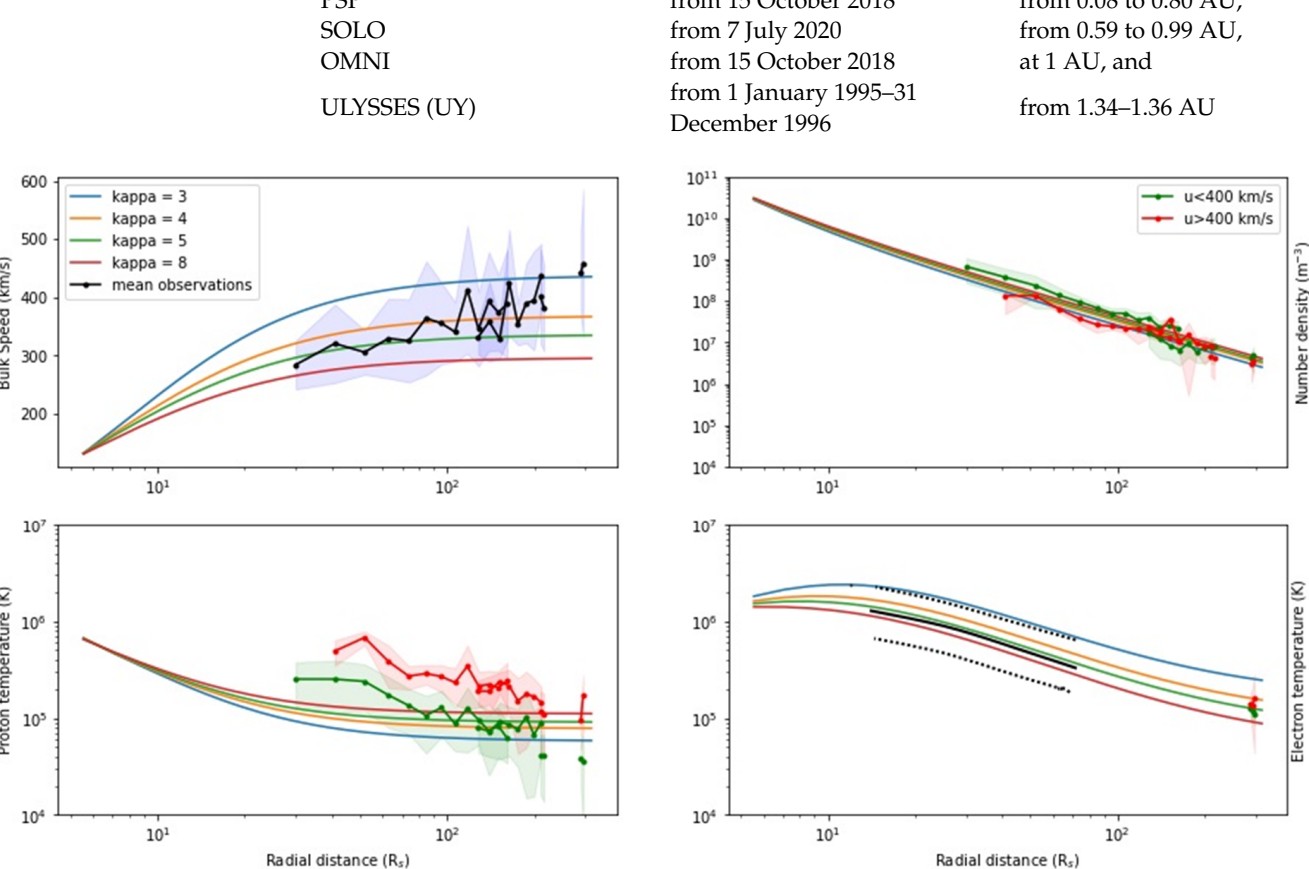

**Figure 2.** Comparison of the data of PSP (17–172 Rs), OMNI (1 AU = 215 Rs), SOLO (126–214 Rs) and UY (288–292 Rs) with the results of the kinetic model for $\kappa$ = 3, 4, 5, 8. The initial parameters used in the model are an initial temperature for the proton of 1.6 $10^6$ K, for the electron 1.4 $10^6$ K, and an exobase altitude of 4 $R_S$. The color black refers to the whole data set, while the green and red colors refer to the data with a speed below and above 400 km/s, respectively. The colored area correspond to the standard deviation. For the electron temperature, the black line corresponds to the fit of averaged observations of PSP between 13 and 60 Rs [39] with the standard deviation shown by the dotted line.

OMNI corresponds to the data observed at the Lagrangian point close to the Earth at 1 AU. ULYSSES observations are older but give observations above 1 AU selected here at latitudes between $-15°$ and $15°$ to be located at low latitudes like for PSP and SOLO. Fast wind is not often observed since coronal holes are rarely observed at low latitudes during minimum solar activity [35]. The minimum solar activity appeared in 2019 and 2020, corresponding to the first orbits of PSP and SOLO analyzed here.

The points represent the average of the data obtained at the corresponding distance, and the colored area represents the standard deviation. The color black refers to the whole data set, while the colors green and red refer to the data with a speed below and above 400 km/s, respectively. In situ data for the other quantities calculated by the model were not provided by the different missions. An exobase at 4 Rs has been assumed. An electron temperature of $1.4\ 10^6$ K and a proton temperature of $1.6\ 10^6$ K were chosen. The observed density profiles correspond very well to the model. The averaged observed velocity is very variable, showing that it would be useful to separate slow and fast winds. The mean velocity corresponds to kappa values between 3 and 8, not to kappa close to 2 since fast wind streams are not frequent at these low latitudes in comparison to the slow wind. The proton temperature decreases faster in the model, but corresponds to the averaged observations at 1 AU. The electron temperature profile obtained with the exospheric model is in excellent agreement with the averaged PSP electron temperature (black line) decreasing as $\sim R^{-0.66}$ obtained between 13 and 60 Rs with quasi-thermal noise spectroscopy [39]. In the model, when the same exobase is assumed, the bulk velocity depends mainly on the kappa index and the bulk speed is then correlated with the electron temperature and anti-correlated with the proton temperature, while the contrary is observed in the solar wind [26]. This suggests that the height of the exobase should not be considered the same for the fast and slow wind, as shown in the next section.

### 3.2. Separating Slow and Fast Wind by Considering Different Exobases

To correctly represent the fast wind, an exobase at lower radii has to be considered since the density is lower in the coronal holes. Figure 3 illustrates what the model can produce by setting the parameters found in Table 1 constrained by PSP, SOLO, and Omni averaged data separated in fast and slow wind.

**Table 1.** Parameters used in the exospheric model to best reproduce the Slow (SSW) and Fast Solar Wind (FSW) as observed on average by PSP, SOLO, and Omni up to 1 AU.

| Wind Type | Exobase Level ($r_0$) | Density at Exobase ($n_0$) | Temperature of Electrons at Exobase ($T_{0e}$) | Temperature of Protons at Exobase ($T_{0p}$) | Kappa (for Electrons) |
|---|---|---|---|---|---|
| SSW | 2.7 Rs | $4 \times 10^{11}$ m$^{-3}$ | $1.5 \times 10^6$ K | $1.25 \times 10^6$ K | 5 |
| FSW | 1.25 Rs | $1 \times 10^{12}$ m$^{-3}$ | $1.35 \times 10^6$ K | $4.06 \times 10^6$ K | 2.23 |

The PSP mean is obtained by binning every 4 Rs from the minimum value and by grouping data above (FSW for fast solar wind) or under (SSW for slow solar wind) 500 km/s. The same method was applied to the SOLO data, where FSW was considered above 600 km/s and SSW under 500 km/s to better differentiate the fast and slow wind at these larger radial distances. The Omni average at 1AU is obtained by computing the mean of all available data relative to the bulk velocity and density of protons. The difficulty remains in distinguishing the FSW from the SSW at a close distance to the Sun since the speed criterion is not unequivocally defined and the available data are limited.

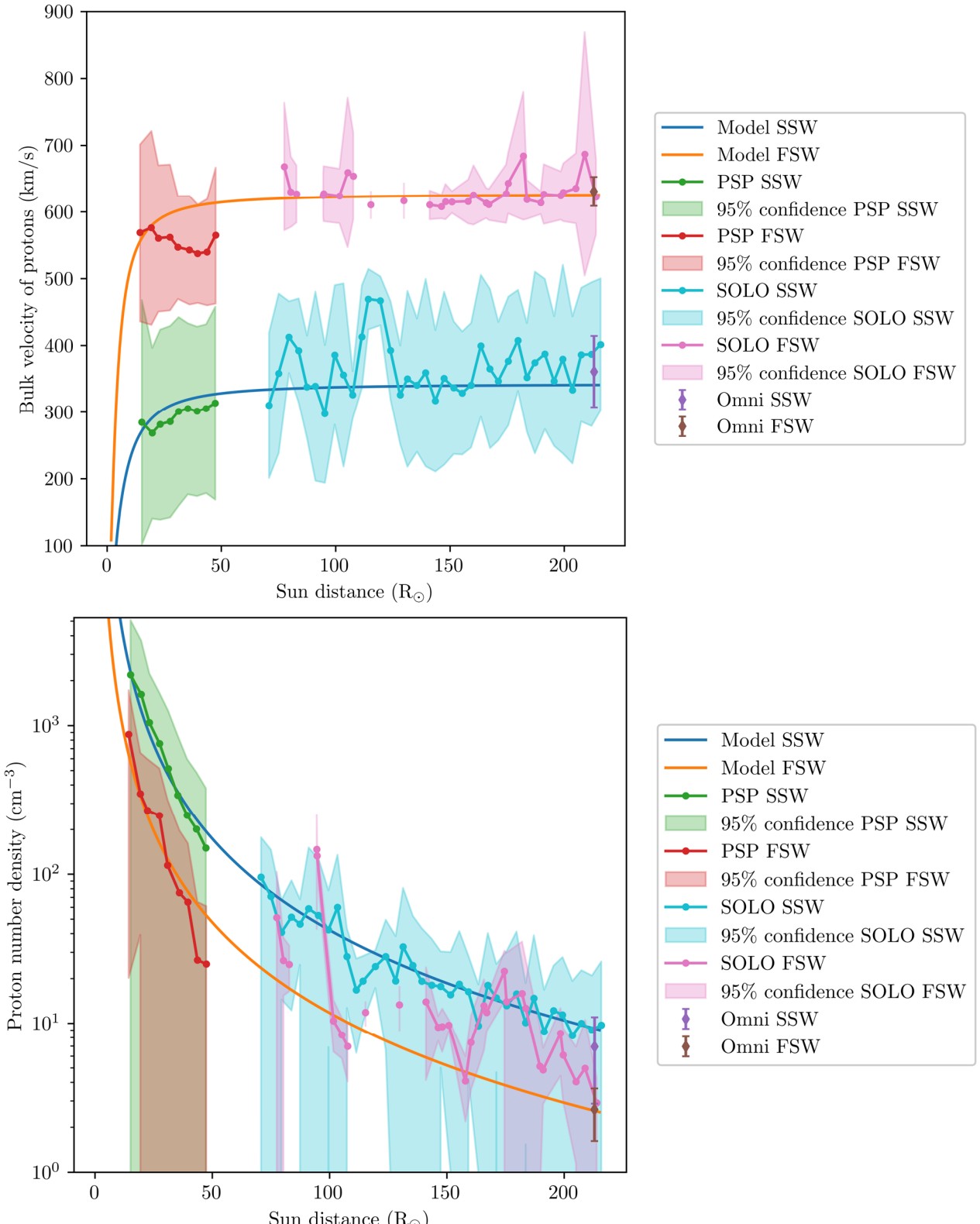

**Figure 3.** Bulk velocity (**top panel**) and density (**bottom panel**) of protons as a function of the solar distance for the exospheric model constrained based on PSP (<50 Rs), SOLO (>70 Rs), and Omni (215 Rs) averaged data for fast solar wind (FSW in orange) and slow solar wind (SSW in blue). The parameters used for the exospheric model can be found in Table 1.

For the fast wind, lower kappa and lower exobase are obtained with the model, as shown in Table 1. For the average fast wind, kappa is found to be slightly higher than 2. To obtain even higher velocities, as regularly observed, lower kappa and/or lower exobase should be assumed. The temperature of electrons also influences the velocity, and it is chosen higher in the slow than in the fast wind to take into account the anticorrelation between solar wind speed and inferred coronal temperature of the Strahl observed with PSP, and which is also consistent with freeze-in temperatures from minor ion ratios [25].

## 4. The Influence of the Electric Potential

As shown previously, the electric potential plays an important role in the acceleration of the solar wind. The electric potential difference between its value at the exobase and its value at a large distance is higher in the fast wind than in the slow wind to accelerate the wind to higher bulk velocities. Recently, Halekas et al. [40] observed with PSP measurements an electric potential that is higher in the slow wind than in the fast wind, leading to an apparent contradiction. However, the observations are in good agreement with the exospheric model, as illustrated clearly in Figure 4 where we use two different exobases, one located at $r_0$ = 1.1 Rs with Kappa = 3.5 for electrons (black), and another at $r_0$ = 6 Rs for Kappa = 3.5 for electrons (red).

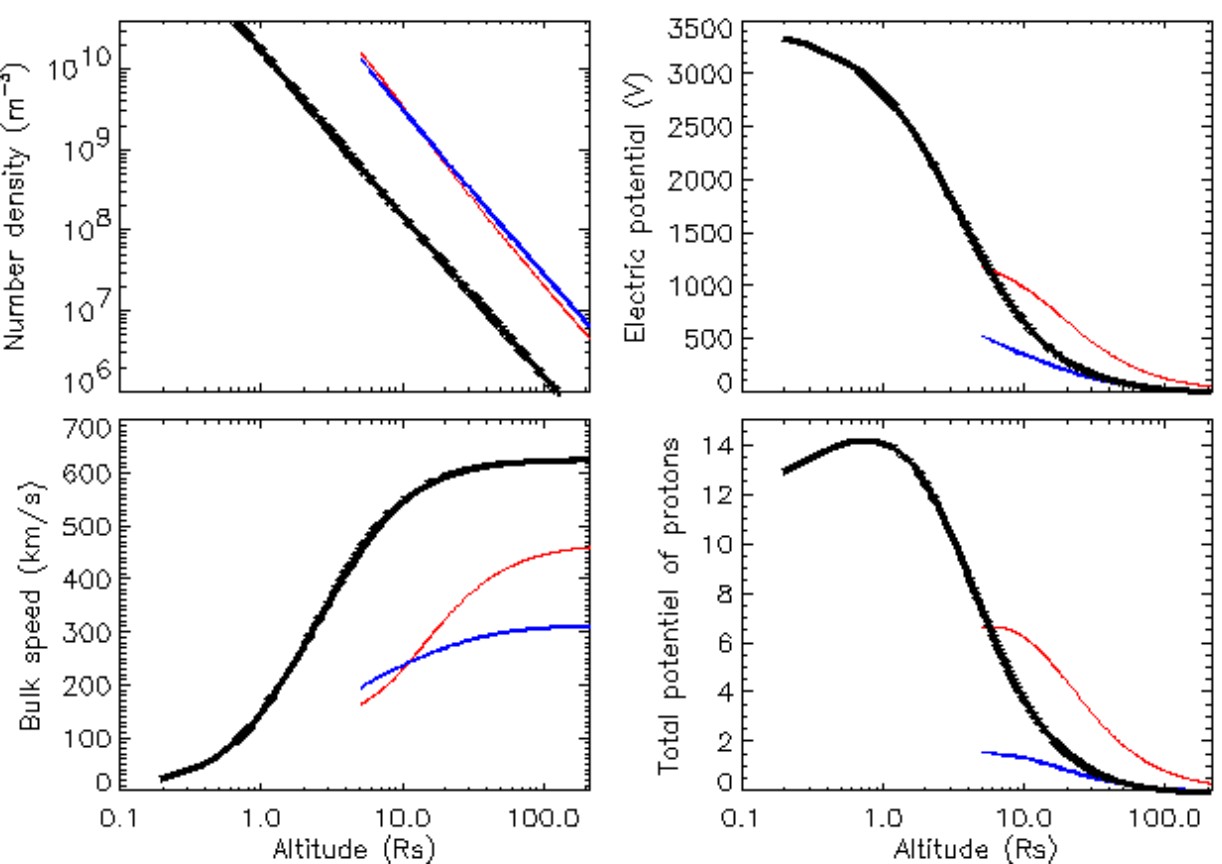

**Figure 4.** Profiles of number density (**upper left**), electric potential (**upper right**), bulk speed (**bottom left**) and total potential of the protons (**bottom right**) obtained with the exospheric model using a temperature of $10^6$ K for electrons and protons at the exobase, respectively located at $r_0$ = 1.1 Rs with $\kappa$ = 3.5 for electrons (black), at $r_0$ = 6 Rs for $\kappa$ = 3.5 for electrons (red) and a Maxwellian for the electrons (blue).

The fast wind and slow wind are mainly differentiated by a lower exobase when the wind originates from the coronal holes where the density is lower than in the other regions of the corona. Starting from a lower exobase, the acceleration starts from lower altitudes with a higher potential difference, but this does not mean that the measured electric potential is lower in the slow wind, especially at distances larger than 10 Rs. This is illustrated in Figure 4, where one can see that the electric potential (upper right panel) is higher for the slower wind in red (see bottom left panel u ~ 460 km/s at large distances) than for the fast wind in black (u ~ 635 km/s) corresponding to the lower exobase.

This is not true for all Kappa values. If we consider a Maxwellian distribution at the higher exobase (blue), the potential is lower, as well as the velocity (u ~ 310 km/s), but it nevertheless also reaches supersonic velocities. As explained previously, Kappa suprathermal tails in the electron distribution are not necessary to form supersonic wind, but they amplify the acceleration. The difference between the slow and fast wind is mainly due to their different exobase levels, associated with lower collisions in the coronal holes where the density is lower. The main parameters that can accelerate the wind are: small kappa index, low exobase, and high electron temperature. All these conditions lead to a higher electric potential difference that accelerates the wind. Nevertheless, the chosen parameters must lead to realistic values of all the moments at all radial distances.

It is worth noting that the total potential (electric and gravitational) of the protons (see bottom right panel) is increasing at low distances and decreasing above a radial distance Rmax [41] located at 2 Rs (thus altitude of 1 Rs) for the black curve for instance. When the exobase is low, the protons are first attracted to the Sun by gravity before the electric force becomes dominant and pushes the protons and other positive ions to compensate for the electron flux. The electric potential ensures the quasi-neutrality and the zero-current conditions. For an exobase above 6 Rs and a high kappa, the total potential of the protons decreases monotonically.

## 5. Electron Distributions

### 5.1. Electron Distribution Obtained with the Exospheric Model

The exospheric model provides not only the moments of the particles but also the distributions of the particles at any radial distance if they have been specified at the exobase. The radial evolution of the electron distribution in the SSW exospheric model is illustrated in Figure 5 for four different radial distances: 2.8 Rs, 3.5 Rs, 4.2 Rs, and 6.1 Rs. The distributions are presented as a function of the perpendicular velocity (y-axis) and the parallel velocity (x-axis) to the Interplanetary Magnetic Field (IMF). One can see that the electron distribution is truncated at the exobase. Indeed, the electrons that have enough energy to escape (escaping electrons above the escape velocity) do not come back, so there are no energetic electrons with a negative velocity (no incoming electrons at all or a very low density that can be considered in the model if some interactions are allowed) (see [8] for all the particle trajectories). On the contrary, low-energy electrons cannot escape because they are attracted toward the Sun by gravity and electric force, so they come back to the Sun and form an isotropic population (ballistic electrons). The conservation of energy gives a lower escape velocity at higher altitudes so that the ballistic core population is reduced. The conservation of the magnetic moment focuses on the energetic escaping electrons of the distribution in the parallel direction. Similar results are obtained for the FSW at even lower radial distances. Below the exobase, it can be assumed that the distributions are isotropic and possibly already shaped as Kappa distributions, which could then explain the heating of the corona ([7,10] for a review).

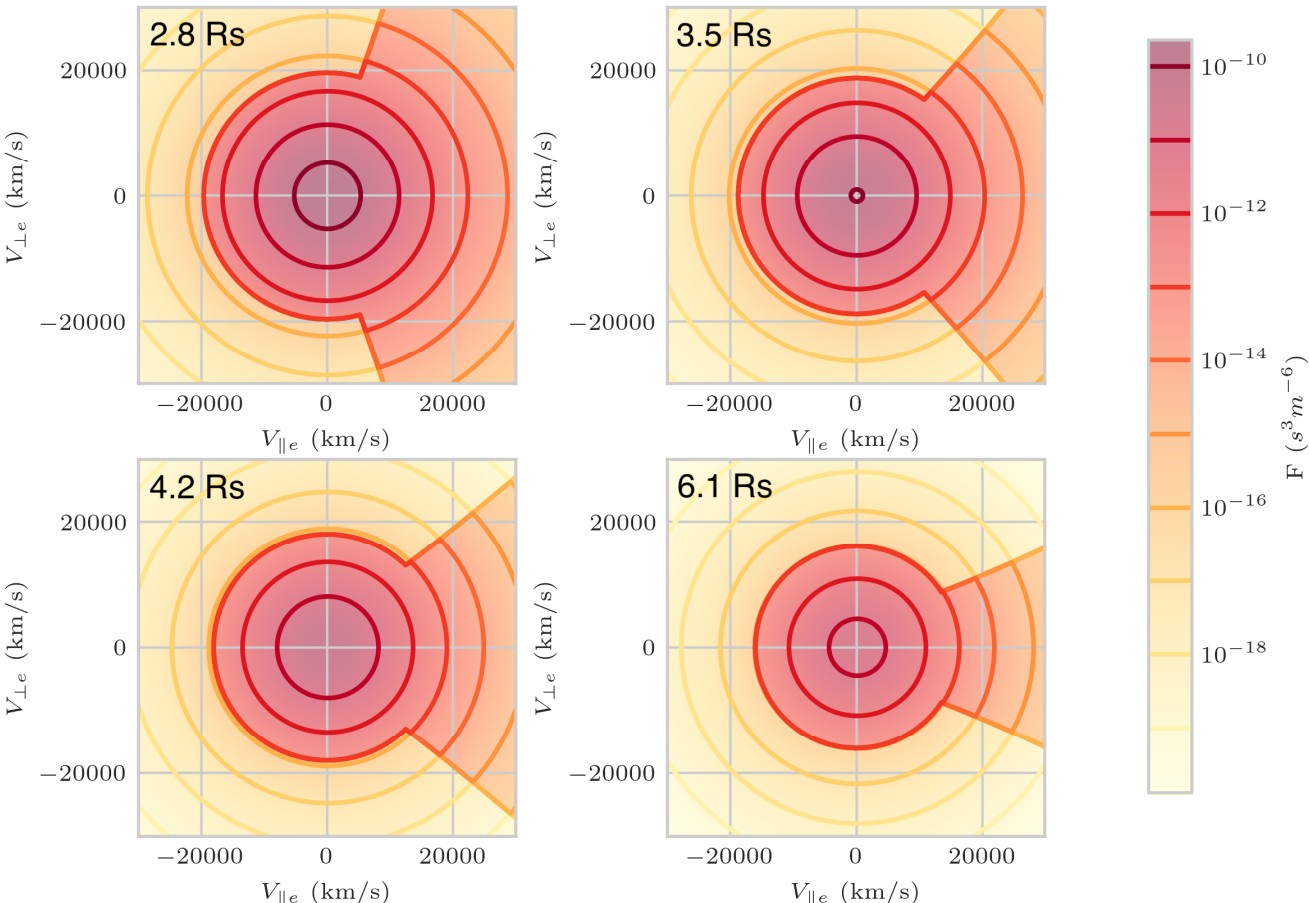

**Figure 5.** Electron velocity distribution obtained in the SSW exospheric model in the velocity plane parallel and perpendicular with respect to the IMF at four different distances: 2.8 Rs close to the exobase (**top left panel**), 3.5 Rs (**top right**), 4.2 Rs (**bottom left**) and 6.1 Rs (**bottom right**).

*5.2. Electron Distribution Observed by PSP in the Velocity Plane*

New observations of the velocity distribution functions of the particles close to the Sun are now accessible with the mission Parker Solar Probe (PSP) launched by NASA on 12 August 2018. PSP completed its 14th perihelion in December 2022. It travels below the orbit of Mercury and will reach distances below 10 Rs in 2024. The Solar Probe ANalyzers (SPAN) sensor onboard PSP measures in situ the logarithmic flux log10 *f* and the pitch angle as a function of the energy (eV) for the ions and the electrons of the solar wind. The SPAN instruments (SPAN-E for electrons and SPAN-Ion for the ions) are electrostatic analyzers that take part in the Solar Wind Electrons Alphas and Protons (SWEAP) Investigation. A description of the instrument can be found in [42] and specifically for SPAN-E in [43] and for SPAN-Ion in [44].

A typical electron distribution observed by PSP at 17.2 Rs is shown in the left panel of Figure 6. In the spacecraft frame, energies for the electrons are measured between 2–2000 eV. The measured values correspond only to the positive perpendicular velocities. The negative ones are obtained assuming perpendicular symmetry. A slight anisotropy is visible in the direction parallel to the magnetic field that is even clearer in 1D (see next section).

The right panel shows the electron distribution found by the exospheric model at 17.2 Rs using the conditions of SSW (slow wind) of Table 1, plotted with the same color scale for direct comparison with the observations. At 17.2 Rs, a very high anisotropy of the electron VDF is found in the model due to the conservation of the magnetic moment. The distribution has a similar density than in the PSP observations. The truncation of the modeled VDF is nevertheless too sharp in comparison to the observations.

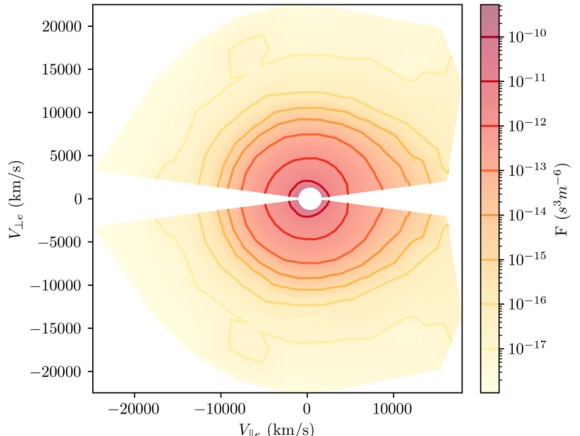 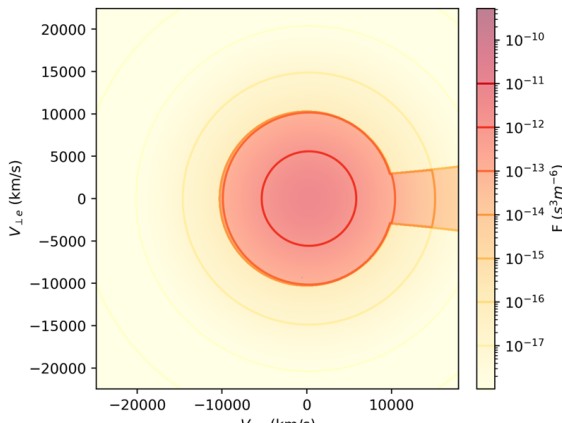

**Figure 6.** (**Left panel**): Electron velocity distribution at a small distance (17.2 Rs) observed by PSP as a function of parallel (x-axis) and perpendicular velocity (y-axis) with respect to the IMF. (**Right panel**): Electron distribution found by the exospheric model at 17.2 Rs using the conditions of SSW of Table 1.

### 5.3. One Dimension (1D) Single Fit for the Strahl and the Halo

Figure 7 illustrates the typical electron velocity distribution measured by PSP at 17.2 Rs in 1D in different directions. The top panels correspond to the Strahl (at a small pitch angle of 7.5°, close to 0°), and the bottom panels to the halo (pitch angle of 82.5°, close to 90°). The VDF of the electrons using all the measured data (black dots in left panels) is fitted by a single Kappa (blue line). In the direction parallel to the magnetic field (top left panel, direction opposed to the Sun), the suprathermal component corresponds to the Strahl (and thus, for the model to the escaping electrons). In the direction perpendicular to the IMF (bottom panels), the suprathermal electrons correspond to the halo (and, for the model, to the transverse trapped electrons). The fitting parameters are given in Table 2.

**Table 2.** Fitting parameters of the PSP electron distribution at 17.2 Rs were obtained with a single Kappa (4 first lines) for the whole velocity range of the observed distribution. The six next lines correspond to a fit with a single Kappa when the low velocities (<3250 km/s) and the very high velocities are not considered due to contamination (see text). When the kappa index is larger than 90, the Maxwellian fit parameters are also provided.

| Angle | | n [$10^9$ m$^{-3}$] (err) | T$_\kappa$ [$10^5$ K] (err) | $\kappa$ | R$^2$ |
|---|---|---|---|---|---|
| Full dataset | | | | | |
| 7.5° | | 4.07 (0.82) | 2.86 (0.29) | 2.16 (0.36) | 0.961 |
| 22.5° | | 3.92 (0.75) | 2.55 (0.31) | 3.49 (0.51) | 0.969 |
| 82.5° | | 4.24 (0.61) | 1.76 (0.14) | 4.10 (0.31) | 0.973 |
| 172.5° | | 3.82 (0.90) | 2.39 (0.30) | 4.79 (0.51) | 0.978 |
| Restricted dataset | | | | | |
| 7.5° | | 1.75 (0.31) | 5.54 (0.63) | 6.41 (2.62) | 0.94 |
| 22.5 | | 1.56 (0.05) | 6.08 (0.13) | 9.22 (1.66) | 0.999 |
| 82.5° | | 1.50 (0.15) | 4.29 (0.23) | 104.3 | 0.996 |
| 82.5° | Maxw | 1.63 (0.10) | 4.32 (0.07) | | 0.997 |
| 172.5° | | 1.56 (0.25) | 4.75 (0.48) | 91.8 | 0.979 |
| 172.5° | Maxw | 2.04 (0.19) | 4.56 (0.14) | | 0.99 |

The fit of the Strahl is most interesting for the model since it corresponds to the escaping particles. When all the data measured at 7.5° are used (top left panel), the $\kappa$ index obtained for the Kappa fit is close to 2, indicating an important presence of suprathermal electrons even close to the Sun. The fit underevaluates the observed data below 1500 km/s, but these low-energy observations are not reliable due to contamination. For the halo

(bottom left panel), the kappa index is larger (κ~4) but remains low, indicating tails also in the perpendicular direction.

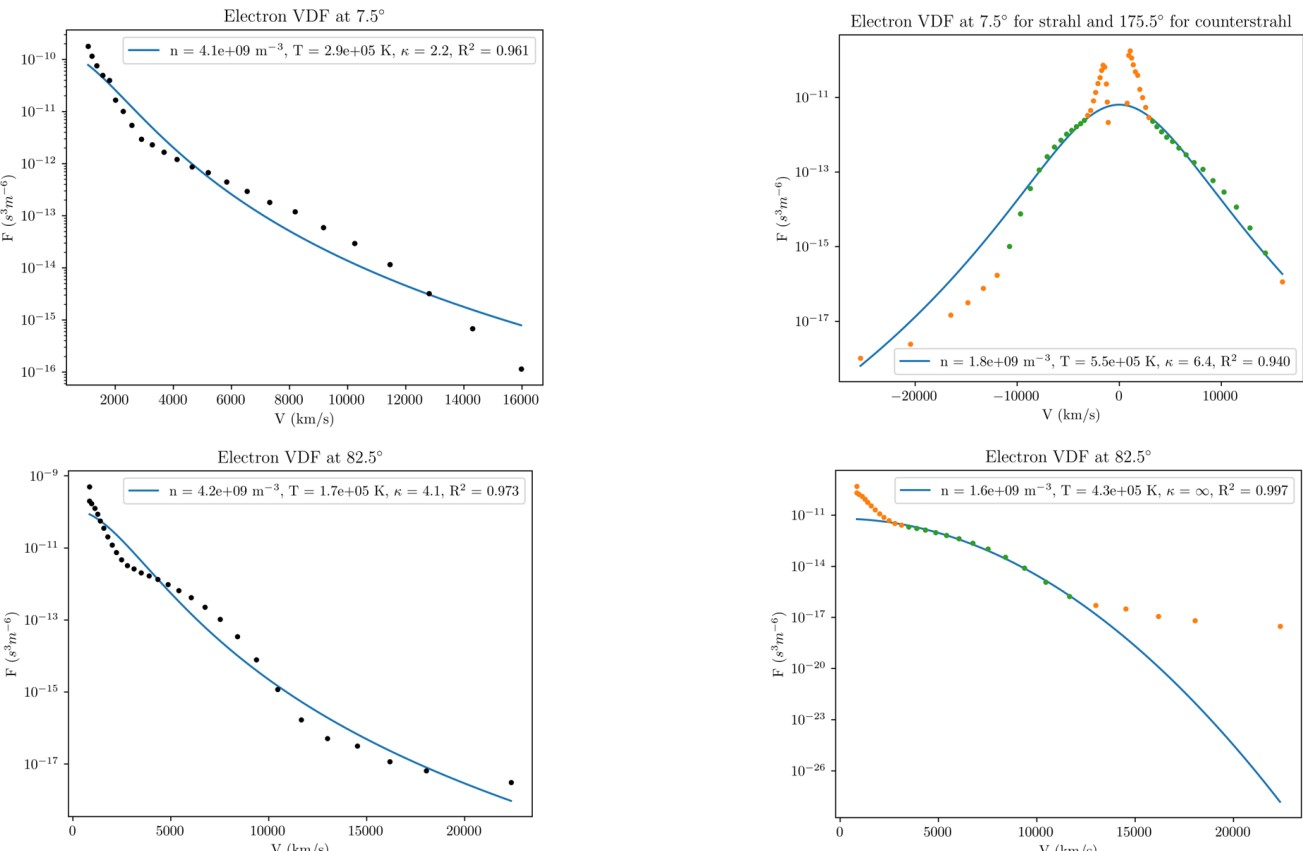

**Figure 7.** PSP observations at 17.2 Rs (dots) fitted by a single Kappa (solid line). (**Top panels**): Strahl (low pitch angle). (**Left**): Fit (in blue) using all the measured points (in black). (**Right**): Fit using only the green points of the Strahl and excluding the orange data points below 30 eV (~3250 km/s) due to secondary contamination. The counterStrahl is also illustrated. (**Bottom panels**): Halo (Pitch angles close to 90°). (**Left**): Fit using all the measured points. (**Right**): Fit using only the green data points and excluding the orange data points below 30 eV (~3250 km/s) and those above 400 eV (~11,900 km/s) due to contamination.

Nevertheless, data points below 30 eV (~3250 km/s) are affected by secondary contamination due to photoelectrons [6,26,38,40]. It is better to exclude them from the fit (see right panels). In such a case, the obtained kappa index is higher but still around 6 for the Strahl, indicating the presence of significant suprathermal tails. In the Strahl direction, the observations go until about 700 eV (~15,700 km/s) and are all valid above 3250 km/s. For the non-Strahl angles (halo and counterStrahl), data points above 400 eV (~11,900 km/s) are dominated by the sensor background. Given the low geometric factor of the sensors needed to avoid saturation near the Sun [43], the instrumental background due to the combination of galactic cosmic rays and natural radioactivity in the microchannel plate detector is more significant than at 1 AU, leading to the ~1/E^2 spectrum seen at higher energies in the non-Strahl angles (see, e.g., [40]). Excluding also these points for the fits in the perpendicular and anti-Strahl direction leads to almost Maxwellian distributions in these directions. The suprathermal "superhalo" electron component that was discovered by WIND and dominates above ~2 keV with a nearly isotropic angular distribution [45] is not accessible to the PSP measurement energy range.

The energy range on which the measurements are obtained is very important for the fit parameters, and it is more limited with PSP than, for instance, with WIND measured at 1 AU, so the results have to be cautiously evaluated when considering the evolution of the tails with the distance [46].

The fitting parameters are highly dependent on the data measurements that are considered in the fit. Table 2 gives the parameters of the electron fits with a single Kappa for different directions, on all the velocities (4 first lines), and on restricted velocity range to exclude unusable data (6 last lines). In parenthesis, the uncertainty on the parameters is given using the standard error.

Using all the observations, the Kappa single fits have lower values of $\kappa$ to be able to represent the entire non-thermal population. This fitted index $\kappa$ has reached its minimum value of 2.16 (+/−0.36) in the direction of the Strahl. In this direction, a higher temperature is also found.

Using only the validated data on the restricted velocity range, the values of $\kappa$ are much higher. The six last lines correspond to this fit with a single Kappa when the very low energy secondary observations (<3250 km/s) and the very energetic points due to background are not considered. The density is then found to be lower (with a factor > 2) than with the full measurements, while the temperature is, on the contrary, much higher (also a factor slightly > 2). The temperature is larger with the restricted velocity range, especially in the Strahl direction. The kappa index becomes much larger but remains significantly close to the parallel direction (Strahl). On the contrary, for the halo and counterStrahl, the kappa index and the error on this parameter become extremely high, showing that a Maxwellian fit is even more appropriate. When the kappa index is larger than 90, the tails are not decreasing as a power law anymore, and the Maxwellian fit parameters are then also provided. Note that for $\kappa$ > 15, the Kappa distribution is already very close to a Maxwellian. The Maxwellian fits give similar values for the density and the temperature as with a Kappa. Note that the number density found by the fit also corresponds to that of the protons observed simultaneously (see next section) at the same distance.

Following a similar procedure for the PSP data, Halekas et al. [26] fitted a Maxwellian to the low energy secondary electron population to remove it and Kappa to the validated observations. The kappa fits of [26] give results similar to those of the right panels of Figure 7 for the valid data.

It should be emphasized that the presence of Strahl in the VDF accelerates the wind in exospheric models. What contributes to the wind is the anisotropy between the flux escaping from the Sun in the antisunward direction and the deficit of particles in the sunward direction. The observed counterStrahl observed by PSP is indeed marked by such a deficit corresponding to the absence or very low contribution of incoming particles, in agreement with the exospheric approach. This deficit was already observed with PSP electron observations at 30 Rs [25] and is here confirmed at even lower radial distances.

Stverak et al. [47] showed with Helios, Cluster, and Ulysses data that better results for the fits are obtained using a sum of two functions: a bi-Maxwellian for the core + bi-Kappa for the suprathermal tails. They found the low-energy core of the velocity distributions is almost Maxwellian, with a core/halo breakpoint around 7 $k_B T_c$ (where $k_B$ = 1.38 $10^{-23}$ J/K is the Boltzmann parameter and $T_c$ is the core temperature) or about two times the thermal velocity. The kappa index of the halo is then found to decrease from 7.8 on average (7.4 in the fast wind) at 0.4 AU to 2.5 at 4 AU [48]. Using similar fits of Maxwellian for the core and Kappa for the halo with PSP observations, Abraham et al. [6] found kappa indices around 4 at very low distances of 0.13 AU (~28 Rs), slightly increasing with the distance, reaching a maximum around 11 at 0.25 AU (~54 Rs) and decreasing to 6 at 0.5 AU (~107 Rs) (see their Figure 2 panel bottom right).

Relations between single Kappa fits and double function fits on the same velocity range were obtained in [49]. In [47], the core was fitted by a bi-Maxwellian, while the suprathermal tails were fitted by a modified bi-Kappa VDF. As shown in [30], the expression of a modified Kappa differs from that of the Olbertian Kappa by a factor ($\kappa$-3/2) and leads

to a temperature independent of κ in the modified Kappa. The fitting parameters are thus not only dependent on the considered velocity range of the data but also on the distributions used for the fits, as already mentioned in [49]. The density and kappa index are nevertheless almost not affected by the use of a modified Kappa or an Olbertian for the fits. Only the temperature is different, with a lower temperature obtained for the Olbertian.

## 6. Proton Distribution Observed by Parker Solar Probe

### 6.1. Proton Distribution in the Velocity Plane

Figure 8 (top panel) illustrates the proton distribution measured by PSP at 17.2 Rs as a function of the perpendicular velocity and the parallel velocity to the IMF. The proton VDF is measured at the exact same time and same distance as the electron VDF of Figure 6 (left panel).

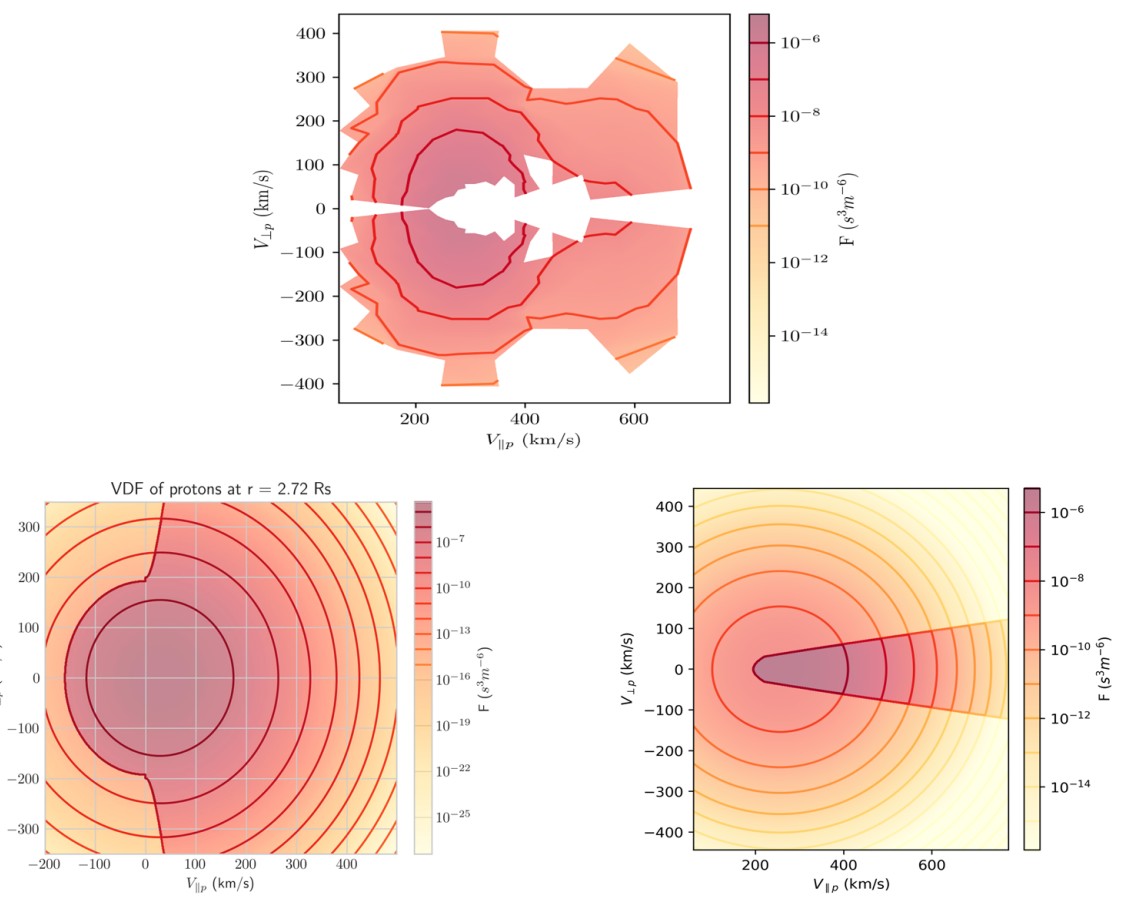

**Figure 8.** (**Top panel**): Proton velocity distribution at a small distance (17.2 Rs) observed by PSP as a function of parallel (x-axis) and perpendicular velocity (y-axis) with respect to the IMF (simultaneously to the electron VDF). (**Bottom panels**): (**left**) Proton VDF assumed by the SSW model just above the exobase at 2.72 Rs and (**right**) found by the exospheric model at 17.2 Rs using the conditions of SSW of Table 1 and assuming a Maxwellian $10^{-4}$ times lower than the whole VDF as background.

The bulk velocity from the VDF of PSP is 299 km/s, and the measured energies for the ions are between 11.67 and 1629.53 eV. Unlike in the case of the electron VDF, the transformation from the instrument/spacecraft frame to the plasma frame is quite significant. Furthermore, the PSP VDF does not cover the low energies in the plasma frame because the distribution is shifted from larger energy in the spacecraft frame, where the instrumental resolution is coarse, to a grid with finer resolution.

In any case, Figure 8 clearly shows the presence of two different populations, a core with $T_\perp > T_{//}$ (contrary to the anisotropy of the full VDF) and a tail in the direction parallel

to the magnetic field. Such antisunward beams were already observed by Helios in the seventies as low as the orbit of Mercury [50] but never so close to the Sun.

The bottom panels show the proton VDF of the exospheric model just above the exobase at 2.72 Rs (left panel) and at 17.2 Rs (right panel) using the conditions of SSW of Table 1. At the exobase, the VDF is also truncated due to the absence of incoming protons. The absence of incoming protons at the exobase leads to $T_{p\perp} > T_{p//}$ proton temperature anisotropies at low radial distances before an inversion occurs after several solar radii in agreement with observations [51]. At 17.2 Rs, the distribution is highly focused in the direction parallel to the magnetic field by conservation of the magnetic moment. The conservation of energy leads to the reduction of the density. The model gives a tail in the parallel direction to the magnetic field, as observed in the solar wind. In 70% of the proton VDF observed by Helios, a proton beam (second peak) was even visible, which is not obtained with the exospheric model. The formation of the proton beam has been suggested to be related to non-uniform solar wind turbulence [52]. The PSP observed VDF has a density close to the average density value expected at this distance (see top right panel of Figure 2). The measured proton density corresponds to the electron density observed simultaneously.

### 6.2. Proton Distribution Fitted by a Maxwellian

Figure 9 represents the measured ion distribution at 17.2 Rs and the fits obtained in a quasi-parallel (top panel) and quasi-perpendicular direction (bottom panel) to the IMF. The black dots correspond to the distribution observed by PSP, and the blue lines represent the best fit with a Maxwellian distribution. Taking each direction separately, in the halo, Strahl, and counterStrahl direction, the distribution of the protons is well-fitted by a Maxwellian distribution. The distribution has a tail in the direction parallel to the IMF, but it does not decrease as a power law contrary to the electrons. This observation justifies that Maxwellians are indeed used in the exospheric model for the protons and not Kappa for the electrons.

The fit parameters are indicated in each panel of Figure, for the counterStrahl and Strahl in the first panel. The temperature is higher for the Strahl than for the halo, while the counterStrahl has the lowest temperature. Additionally, a clear deficit is visible in the sunward direction, in good agreement with exospheric assumptions. The densities are very different following the direction (2.7 times larger for the halo than for the counterStrahl), due to the lack of measurements at low velocities.

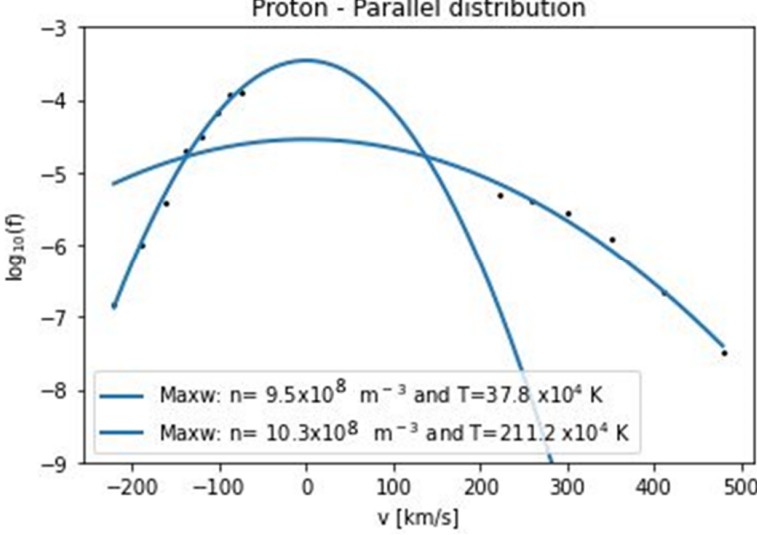

**Figure 9.** *Cont.*

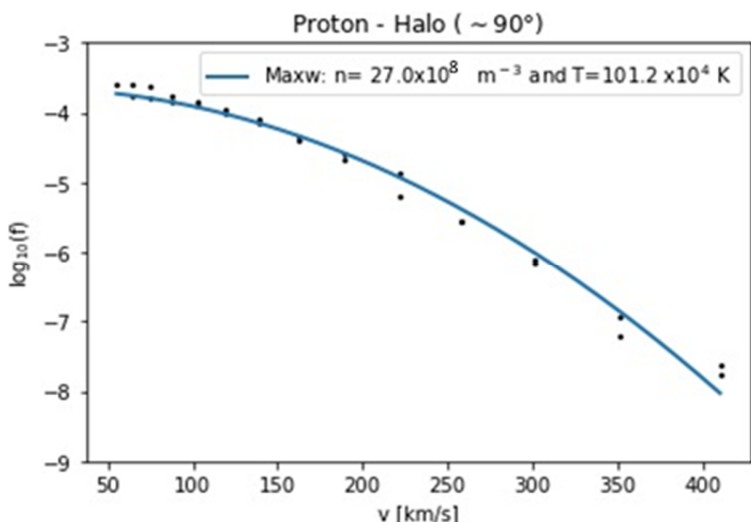

**Figure 9.** Maxwellian fit of PSP proton distributions at 17.2 Rs in $s^3/(cm^3\ km^3)$ for: (**top panel**) the counterStrahl (upper line) and Strahl (lower line that fits the observed positive velocities, with higher temperature) (angle around $0°$, $R^2 = 0.98$) and (**bottom panel**): Halo (angle around $90°$, $R^2 = 0.99$).

## 7. Correlation Proton Bulk Velocity-Temperature

Strong correlations between the bulk velocity and the proton temperature were observed by several spacecraft at different distances ([48] and references there in). The 2 cases of slow and fast wind of the exospheric model reproduce such correlations due to the lower exobase for faster wind. A lower kappa alone leads to higher bulk velocities with slightly lower proton temperatures (see Figure 1 of the present article). It should be noted that associating a lower exobase with faster wind reproduces the velocity/proton temperature correlation. The proton temperatures decrease faster than the PSP observations, but they lead to proton temperatures at 1 AU corresponding to what is obtained on average at 1 AU with OMNI data taken during the same time period: $T_p \sim 5\ 10^4$ K in SSW and $2\ 10^5$ K in FSW.

Figure 10 shows that the correlation observed at large distances is also observed by Parker Solar Probe, even at very low radial distances, here for two ranges of distances: 15–17 Rs and 50–52 Rs. The correlation remains very high throughout the heliosphere. It means that the process that causes this correlation (which is not yet fully understood) takes place at low distances < 15 Rs. The linear regressions are displayed with their associated $R^2$ values. Additionally, the green linear regression ensures that the linear regression at 50–52 Rs is not biased by the addition of data from more encounters (in blue for right-sided figures) since the low distance of 15–17 Rs was not reached during the first encounters.

The different colors indicate the dates of the inward orbit shortly before perihelion or the outward orbit shortly after perihelion. Therefore, we see two groups of particles with approximately the same color since PSP crosses the determined Sun distance ranges (50–52 Rs or 15–17 Rs) two times in one orbit. Figure 10 (right panel) clearly illustrates a cluster of particles associated with fast solar wind above 450 km/s and another under 450 km/s. In Figure 10, left closer to the Sun, it is much harder to distinguish the two clusters of particles.

The proton temperatures are also highly correlated to the alpha temperatures. Figure 11 reveals a high correlation close to the Sun (left panel) and reducing at further distances (right panel). This behavior indicates that the temperature of alphas is reducing faster with increasing distance than the temperature of protons. It should be noted that the data are limited to the available alpha data. Furthermore, the temperature data should be considered with caution because the field of view of SPAN-Ion is partly blocked by the heat shield, especially during times further away from perihelion. During those times, the spacecraft speed is notably in the sunward direction, rendering the core of the VDF

not visible by the instrument and inducing a large uncertainty on temperature, whereas, during perihelion times, the core of the VDF is visible due to the spacecraft speed being essentially perpendicular to the Sun.

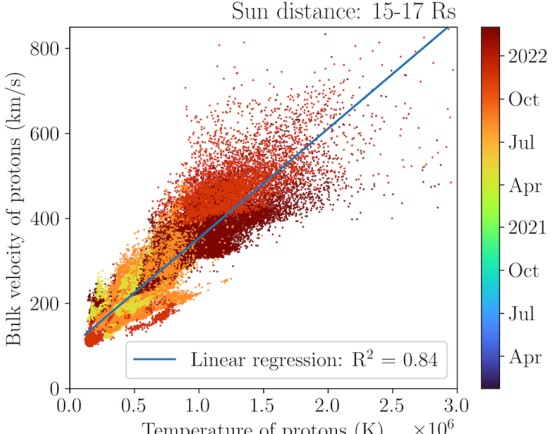 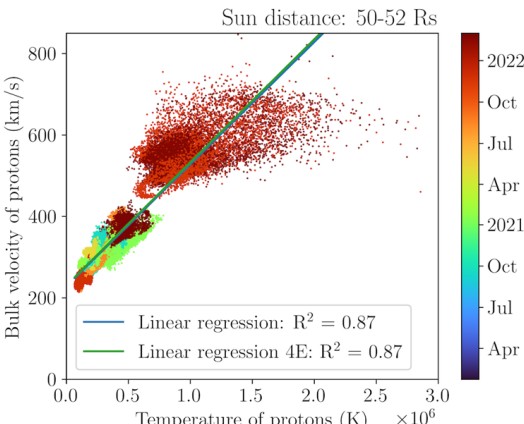

**Figure 10.** Proton bulk velocity as a function of the temperature of Parker Solar Probe observations. (**Left panel**): between 15–17 Rs from 4 orbits from April 2021 to March 2022. (**Right panel**): between 50 to 52 Rs from 6 orbits from September 2020 to March 2022. The linear regressions considering the entire displayed data for each figure are shown in blue, while the green line in the right panel corresponds to the linear regression considering only the four encounters available at 15–17 Rs.

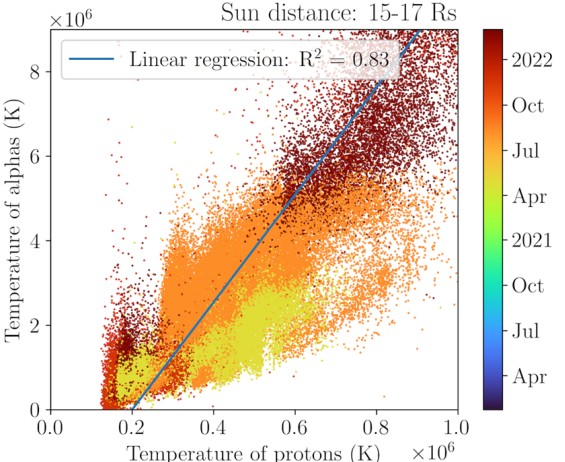 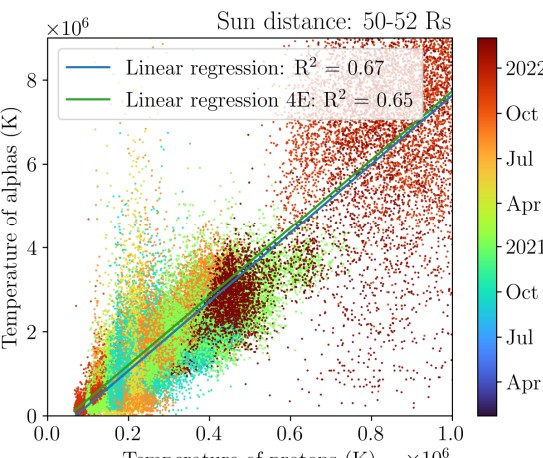

**Figure 11.** Joint occurrence of alpha temperature and proton temperature for Parker Solar Probe observations. (**Left panel**): between 15–17 Rs from 4 encounters from April 2021 to March 2022. (**Right panel**): between 50–52 Rs from 6 encounters from September 2020 to March 2022. The linear regressions considering the entire displayed data for each figure are shown in blue, while the green line corresponds to the linear regression considering only the four encounters available at 15–17 Rs.

Figure 12 illustrates that the correlation between alpha/proton density ratio and proton bulk velocity found at 1 AU [48] is not observed at 15–17 Rs ($R^2 = 0.13$) and is only slightly higher at 50–52 Rs ($R^2 = 0.23$). High ratios larger than 10% are only observed in the fast wind with v > 500 km/s, but the fast wind can also have low ratios. The same behavior is found for the correlation between alpha/proton density ratio and proton temperature, which seems to indicate that both of those correlations are produced with solar wind expansion. The very high dispersion in the density ratio observed at 50–52 Rs is quite surprising since previous measurements made at 1 AU indicated a lower abundance variation in the high-speed solar wind than in the slow wind (e.g., [53]). There is no clear Helium abundance associated with SSW or FSW in the PSP observations.

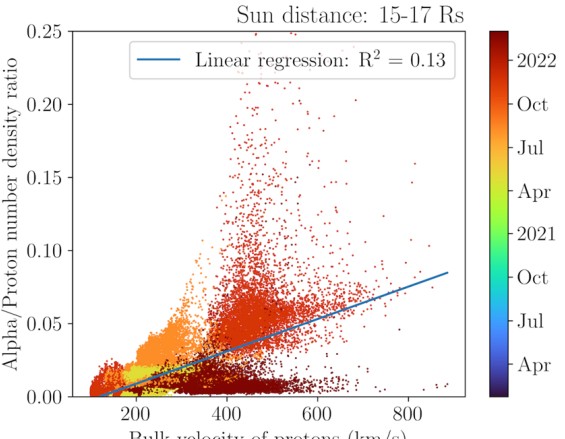 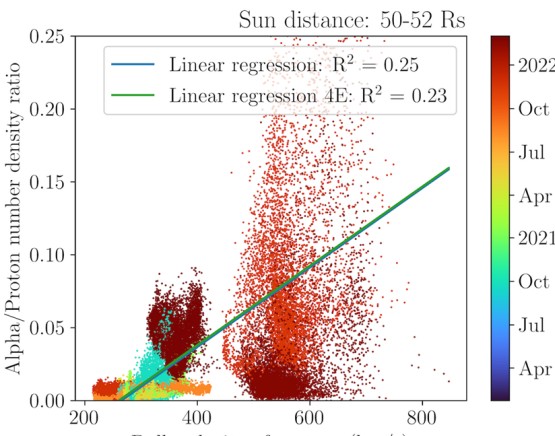

**Figure 12.** Same as Figure 11, but for the alpha/proton number density ratio as a function of the proton bulk velocity for Parker Solar Probe observations between 15–17 Rs for encounters from April 2021 to March 2022 (**left panel**) and between 50–52 Rs for encounters since September 2020 to March 2022 (**right panel**).

The solar wind exospheric model [36] also allows us to include minor ions like alpha particles. The model shows that even if the minor ions are more difficult to accelerate by the electric potential due their higher mass on charge ratios (for instance, the mass of alpha particles is four times higher than that of the protons, while their charge is only two times larger, leading thus to a Rmax higher than for protons), these ions can reach high bulk velocities (even sometimes higher than protons) due to their very high temperatures, that are observed to be much higher than that of the protons at the exobase [12]. These high temperatures can be due to the velocity filtration effect if Kappa distributions are already present below the exobase at the top of the chromosphere for the electrons and the minor ions, as observed at larger distances [9]. The proportion of minor ions and their bulk velocities in the solar wind model depend a lot on the boundary conditions (density, temperatures, kappa) at the exobase, and these conditions are themselves determined by the assumptions made in the chromosphere [9]. Any measurement of the minor ions at low distances from the Sun is thus crucial for the model, and the correlations observed in Figures 11 and 12 give insight into the physical processes appearing in the acceleration region of the solar wind.

## 8. Conclusions

Exospheric models provide a very simplified first approximation because they consider only the effects of the different forces, neglecting Coulomb collisions and wave-particle interactions. Nevertheless, they show already very interesting characteristics corresponding to the observations. They emphasize that the electric potential can accelerate the wind to supersonic velocities, even considering simple Maxwellian distributions for the particles at the exobase. The electric potential measured by PSP seems higher in the slow wind than in the fast wind [40]. This is in good agreement with the exospheric model, as illustrated in this article because a lower exobase in the fast wind ensures the acceleration of the wind at lower radial distances. Moreover, the possible presence of suprathermal particles in the corona has important effects on the plasma temperature increase. An enhanced population of energetic electrons accelerates the solar wind to larger bulk velocities, especially in the case of a low exobase. This gives a natural explanation for the fast wind originating from coronal holes, where the density is lower than in the other coronal regions.

Typical velocity distribution functions of electrons and protons observed by PSP at 17.2 Rs have been fitted by Kappa and Maxwellian functions, and this demonstrated that suprathermal electrons are already present at a low distance from the Sun in the Strahl direction. Tails are observed in the direction parallel to the magnetic field, and they are

reproduced in the model through the conservation of the magnetic moment. A clear deficit in the counterStrahl direction is also measured in the particles VDF, in agreement with the exospheric approach.

Due to the uncertainties in the PSP VDF measurements, the fitting parameters are quite uncertain. Nevertheless, the kappa index seems higher at low altitudes than at larger distances, at least for the halo. This is coherent with the effects of Coulomb collisions and with the results of models that include them [54]. Nevertheless, the electron distributions at low distances already show a Strahl suprathermal Kappa tail. Due to the contamination of the measurements at very low and very high energies, the valid velocity range accessible to PSP is restricted, inducing greater uncertainties on the kappa index. At larger distances, kappa values were more precisely obtained from the fits of the observed distributions and showed values close to or even sometimes lower than 2, justifying the use of a regularized Kappa distribution to improve the model. Such distributions have a cut-off for velocities approaching the speed of light, preventing any divergence of the moments, even for low kappa values. For values of $\kappa > 2$, the exospheric regularized Kappa model keeps the same results as the Kappa model of [36].

The model's simplification, specifically its neglect of Coulomb collisions and wave-particle interactions, leaves room for future enhancements. Future studies that take these factors into account might be able to provide an even more comprehensive model of the solar wind. Moreover, future data of PSP even closer to the Sun and of SOLO during the coming more active period of the Sun should also improve the statistics, especially by providing more observations of the fast wind.

**Author Contributions:** Conceptualization and methodology, V.P.; writing—original draft preparation, V.P.; software, and formal analysis and investigation V.P., M.P.d.B. and C.A.; validation, J.H., R.L. and P.W.; writing—review and editing, V.P., M.P.d.B. and J.H.; funding acquisition, V.P. All authors have read and agreed to the published version of the manuscript.

**Funding:** The project 21GRD02 BIOSPHERE has received funding from the European Partnership on Metrology, co-financed by the European Union's Horizon Europe Research and Innovation Programme and by the Participating States. JSH acknowledges support from the PSP mission through contract NNN06AA01C and the Living with a Star program through NASA grant 80NSSC22K1014.

**Informed Consent Statement:** Not applicable.

**Data Availability Statement:** PSP data are available on https://cdaweb.gsfc.nasa.gov (accessed on 1 March 2023), Solar Orbiter on https://soar.esac.esa.int/soar/ (accessed on 1 March 2023), OMNI on https://omniweb.gsfc.nasa.gov (accessed on 1 March 2023). ULYSSES on https://spdf.gsfc.nasa.gov/pub/data/ulysses (accessed on 1 March 2023). The Kappa exospheric model is available on the CCMC website https://ccmc.gsfc.nasa.gov/ (accessed on 1 March 2023).

**Acknowledgments:** VP and JH thank the International Space Sciences Institute (ISSI) and the participants in 2021–2023 ISSI workshops for the project "Heliospheric energy budget: from kinetic scales to global solar wind dynamics".

**Conflicts of Interest:** The authors declare no conflict of interest.

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
