# Peer review of "Exospheric Solar Wind Model Based on Regularized Kappa Distributions for the Electrons Constrained by Parker Solar Probe Observations"

_plasma, doi:10.3390/plasma6030036_

Round 1

Reviewer 1 Report

The paper presents an enhanced kinetic exospheric model of solar wind that accommodates regularized Kappa distributions for kappa indices lower than 2. It leverages new observations from the Parker Solar Probe to refine the model.

One of the primary strengths of the paper is its rigorous engagement with previous models, providing a solid backdrop for their new findings. This foundation allows them to demonstrate how their model improves upon these previous efforts, offering an innovative interpretation of the role of the electric potential and its impact on wind velocities.

The paper also excels in its application of the Kappa exospheric model, which incorporates regularized Kappa distributions to compute heat flux, without the limitations typically imposed by Olbertian Kappa or modified Kappa. This improvement is clearly demonstrated and well-argued, providing a new and robust way to model the solar wind.

The study also draws on cutting-edge observational data from the Parker Solar Probe, grounding their theoretical modeling work in actual measurements. This reliance on new observational data adds credibility to the study's conclusions and ensures that their models reflect the most up-to-date understanding of solar winds.

One potential limitation of this work is its neglect of Coulomb collisions and wave-particle interactions. While the authors acknowledge this simplification, further exploration of these interactions could lead to a more comprehensive model.

Additionally, the model's dependency on the Parker Solar Probe data could also be a limitation. While the probe provides valuable new insights, it's worth noting that all models are only as good as the data they're based on. Future observations may challenge or refine the understanding put forward in this paper.

In conclusion, this paper makes a substantial contribution to the understanding of solar wind models. By employing regularized Kappa distributions and leveraging new data from the Parker Solar Probe, the authors provide a robust model of the solar wind's characteristics. However, the model's simplification, specifically its neglect of Coulomb collisions and wave-particle interactions, leaves room for future enhancements. Future studies that take these factors into account might be able to provide an even more comprehensive model of the solar wind.

Author Response

The authors thank the reviewer for very positive comments and suggestions to improve the manuscript. 

We fully agree concerning the limitations of the model concerning the Coulomb collisions and wave-particle interactions. We added sentences in the conclusions to point out what could improve comprehensive model of the solar wind in future studies and how the future measurements of PSP and SOLO could bring better statistics, especially for the fast wind.

Reviewer 2 Report

The paper titled “Exospheric Solar Wind Model Based on Regularized Kappa Distributions for the Electrons Constrained by Parker Solar Probe Observations” by Pierrard et al. presents an updated exospheric solar wind model, using regularized Kappa distributions and constraining model parameters with recent Parker Solar Probe (PSP) observations of the near-Sun solar wind.

The paper is reasonably well written and the comparison of the exospheric model with new PSP observations will be of interest to the community. The paper also highlights some interesting properties of the near-Sun solar wind plasma observed by PSP. I have the following comments that I would like the authors to address before further consideration for publication.

Major remarks -

1. The manuscript claims that the exospheric model is improved with regularized Kappa distributions, allowing kappa to be equal to or smaller than two. However, the constrained kappa that results from comparison with PSP observations is greater than two. Further, the comparisons with PSP data for different model kappa values (Figure 2) do not show kappa < 3. Examining Figure 1, it appears that kappa equal to 2 and below will lead to poor agreement between model and PSP observations. Therefore, it seems questionable to me that use of regularized Kappa distributions improves the model. To support this claim, the authors should show comparisons of kappa </= 2 (less than or equal to 2) models and observations.

The authors write on line 165, “For values of κ > 2, the regularized Kappa distributions give the same moments as the standard...”. If kappa </= 2 is not allowed by comparison with data, then the use of regularized kappa distributions seems rather superficial. Unless the authors have strong reason to believe otherwise, I suggest they remove the misleading claim that regularized kappa distributions improve the model until a comparison is made between observations (presumably above 1 AU) and the model with kappa </= 2. Further, it should be mentioned that the model results in Figure 1 do not agree with PSP observations in Figure 2.

Finally, since the authors say that at large distances solar wind observations indicate kappa </= 2, are they making an implicit statement that the kappa parameter describing the solar wind must change with distance from the Sun?

2. Section 2 is lacking clarity. The terms appearing in equation (1) are not explicitly defined (v, u, r). On line 149, is it stated that \alpha is the cut-off parameter, or is it stated that \alpha < the cut-off parameter? What is “n” in equation (2)? Please check Equation (3) for typos, for instance the subscripts “[ ]”, “[0]]”.

3. Since the model shown here does not include alpha particles, what is the point of Figures 11 and 12?

Other remarks -

4. Line 117, typo: “the exospheric of the solar wind”

5. In the caption of Figure 1, what does “raw” mean?

6. Are there any observational constraints on the height of the exobase, beyond the fact that it should be lower for fast wind compared to slow wind?

7. In Figure 3 (top panel), it appears that the FSW model has been more strongly weighted with SolO data compared with PSP. Is this the case?

8. Line 317, typo: “electron flu”.

N/A

Author Response

We thank the reviewer for the careful reading of the manuscript and the useful suggestions that helped us to improve the article.

1.Kappa close and lower than 2 were often observed by ULYSSES, not so much because it was at large distances, but because it flew above the polar regions of the Sun (see [31] for instance). During minimum solar activity, fast solar wind was clearly originating from the high latitude regions, as illustrated by McComas et al. (2008). We added this reference ([35]) and specify in the manuscript that kappa close or lower than 2 corresponds to this fast wind L200. In average for the fast wind close to the equatorial plane (with PSP and SOLO since they remain at low latitudes), we find kappa =2.23 to best reproduce it (see Table 1). This is only average for the fast wind: it means that faster wind sometimes observed can be due to kappa even lower than 2. We specify in the manuscript that it is not the most frequent case (L203). Slow wind is much more frequent in the equatorial plane, especially during minimum solar activity corresponding to the data analyzed up to now with PSP and SOLO. That is why, in Figure 2, when all data are used, the average bulk velocity gives values closer to the slow wind. The separation in slow and fast wind, as shown in Figure 3, shows the best agreement with low kappa value for the fast wind.

  1. Section 2 has been improved by defining the terms appearing in equation (1) (v, u, r, t)

    The parameter alpha is the cutoff parameter limited to: 0 < α < 1. What is “n” in equation (2)? The factor n in equation 2 is the number density, as explained L140.  Please check Equation (3) for typos, for instance the subscripts “[ ]”, “[0]]”. The subscript [ ] is correct (we used the same notations as Lazar et al. 2020 [32]). The double ]] has been corrected.

  2. The model includes also ions, even if the results are not specifically shown in the present paper. The correlations shown for the protons and for the helium ions give important ways to improve the solar wind models.

    We added a paragraph at the end of section 7: The solar wind exospheric model [36] also allows us to include the minor ions like alpha particles. The model shows that even if the minor ions are more difficult to accelerate by the electric potential due their higher mass on charge ratios (for instance, the mass of alpha particles is 4 times higher than that of the protons, while their charge is only 2 times larger), these ions can reach high bulk velocities (even sometimes higher than protons) due to their very high temperatures, that are observed to be much higher than that of the protons at the exobase [12]. These high temperatures can be due to the velocity filtration effect if Kappa distributions are already present below the exobase at the top of the chromosphere for the electrons and the minor ions, as observed at larger distances [9]. The proportion of minor ions and their bulk velocities in the solar wind model depend a lot of the boundary conditions (density, temperatures, kappa) at the exobase and these conditions are themselves determined by the assumptions made in the chromosphere [9]. Any measurement of the minor ions at low distances from the Sun is thus crucial for the model, and the correlations observed in Figures 11 and 12 give insight of the physical processes appearing in the acceleration region of the solar wind.

  3. Thanks for pointing out these typos. They have been corrected in the new version of the manuscript. 
  4. “raw” mean? Replaced by row.
  5. The exobase has to be located in the solar corona, thus higher than 1.1 Rs and correspond to the region where the collisions become negligible.
  6. SOLO has a larger range of distances than PSP, that is why its weight seems higher in the fit.   Also OMNI is important in the fit because many data are available at 1 AU. The PSP observations do not show very high-speed solar wind. At low distances where the solar wind is still accelerating, it is not easy to separate slow and fast winds, as explained in the text (seeL297). To improve the figure 3, we changed the limit to separate slow and fast wind in PSP data to 500 km/s.

Round 2

Reviewer 2 Report

N/A

N/A